# Influence of Spatial Dispersion on the Electromagnetic Properties of Magnetoplasmonic Nanostructures

**DOI:** 10.3390/nano11123297

**Published:** 2021-12-04

**Authors:** Yuri Eremin, Vladimir Lopushenko

**Affiliations:** Department of Computational Mathematics and Cybernetics, Moscow Lomonosov State University, 119991 Moscow, Russia; eremin@cs.msu.ru

**Keywords:** magnetoplasmonics, discrete sources method, spatial dispersion, non-local optical response

## Abstract

Magnetoplasmonics based on composite nanostructures is widely used in many biomedical applications. Nanostructures, consisting of a magnetic core and a gold shell, exhibit plasmonic properties, that allow the concentration of electromagnetic energy in ultra-small volumes when used, for example, in imaging and therapy. Magnetoplasmonic nanostructures have become an indispensable tool in nanomedicine. The gold shell protects the core from oxidation and corrosion, providing a biocompatible platform for tumor imaging and cancer treatment. By adjusting the size of the core and the shell thickness, the maximum energy concentration can be shifted from the ultraviolet to the near infrared, where the depth of light penetration is maximum due to low scattering and absorption by tissues. A decrease in the thickness of the gold shell to several nanometers leads to the appearance of the quantum effect of spatial dispersion in the metal. The presence of the quantum effect can cause both a significant decrease in the level of energy concentration by plasmon particles and a shift of the maxima to the short-wavelength region, thereby reducing the expected therapeutic effect. In this study, to describe the influence of the quantum effect of spatial dispersion, we used the discrete sources method, which incorporates the generalized non-local optical response theory. This approach made it possible to account for the influence of the nonlocal effect on the optical properties of composite nanoparticles, including the impact of the asymmetry of the core-shell structure on the energy characteristics. It was found that taking spatial dispersion into account leads to a decrease in the maximum value of the concentration of electromagnetic energy up to 25%, while the blue shift can reach 15 nm.

## 1. Introduction

Localized surface plasmon resonance (LSPR) is a collective oscillation of conduction electrons at the interface between plasmon metal and dielectric upon incident light excitation. The LSPR causes an enhancement in the local electric field by several orders of magnitude on the interface at a distance less than a wavelength, which makes it possible to concentrate energy in ultra-small volumes. This property finds many practical applications in electromagnetic energy storage and conversion devices, chemistry, biology, nanomedicine, and solar cells [1,2].

Plasmonic core-shell nanoparticles have gained huge popularity compared to homogeneous ones due to their multifunctional properties achieved by manipulating the materials of their core or shell. Nanoshells provide greater flexibility in tuning plasmon resonances over a wide range of wavelength, as well as producing the desired enhancement of electromagnetic fields. In particular, core-shell particles made from composite materials have become valuable for energy storage and conversion, optical amplifiers, surface-enhanced Raman scattering, photothermal enhancement, solar cells, cancer diagnostic and treatment, and plasmonic nanolaser (SPASER) [3,4,5,6,7,8,9], finding various applications in many areas such as industrial, clinical, biological, environmental, and food analysis.

Via the latest advances in materials science, it has now become possible to synthesize nanomaterials with predetermined physicochemical properties, well-defined dimensions, shape, and composition [3,5]. As a result, nanomaterials now emerge as a basis for the further development of nanoplasmonics and nanoengineering [10,11]. Magnetoplasmonics based on composite nanostructures is widely used in numerous biomedical applications. Special attention is paid to the core-shell Fe_n_O_m_@Au nanostructures, which exhibit both plasmonic and magnetic properties and are currently used in optical sensors, electrochemical DNA biosensors, for tumor imaging, and cancer therapy [12,13,14,15]. The ability to control the optical properties of such nanoparticles in a wide spectral range and the adjustable size of the composite make these nanostructures an important object of magnetoplasmonic research [15,16,17].

Magnetoplasmonic nanostructures have become an indispensable tool in nanomedicine due to a number of advantages. This is primarily due to a high refractive index and the possibility of cheap and fast synthesis [11]. The gold shell protects the magnetic core from environmental oxidation and corrosion, thereby providing a biocompatible platform for tumor imaging and cancer treatment. By adjusting the relative size of the core and the shell thickness, the LSPR can be shifted from the ultraviolet to the near infrared range, where the depth of electromagnetic wave penetration is maximum due to low scattering and absorption in human tissues [13]. Thanks to the magnetic core, Fe_n_O_m_@Au nanoparticles can be directed to tumor cells by an external magnetic field, localized there, and used for photothermal therapy. Importantly, photothermal therapy using materials that absorb near infrared light manifests itself as a promising approach for selectively killing tumor cells with less harmful effects in healthy cells, while simultaneously promoting faster recovery [8].

The rapid progress in the synthesis of magnetoplasmonic nanostructures leads to their continuous miniaturization [18,19,20]. Already, Fe_n_O_m_@Au nanostructures can now be synthesized with an average size of 15–25 nm, including a gold shell thickness of 2–5 nm [21,22,23]. With a decrease in the thickness of the plasmon shell to several nanometers, the electron-electron interactions in metals should be taken into account much more accurately. The fact is that when the characteristic size of the metal shell becomes comparable to the Fermi wavelength of electrons in this metal (~5 nm for gold and silver), the so-called spatial dispersion of the metal arises. In this case, the conventional local relations between the electric field and the displacement included in the system of Maxwell’s equations are proved to be insufficient for a rigorous description of the electromagnetic properties, since the quantum effect of spatial dispersion emerges [24,25]. To study such effects, one can use a purely quantum approach based on solving the Schrödinger equation for a cloud of electrons in the metal [26]. However, such an approach becomes computationally expensive for particles larger than tens of nanometers and for metals with a high density of free charge carriers inside, similar to noble metals.

There are more popular approaches that allow one to consider the emerging quantum effects, while remaining within the framework of Maxwell’s electromagnetic theory. One of such approaches, accounting for the arising spatial dispersion of the plasmonic material, is the Drude hydrodynamic model [27,28] and its modifications, which are applicable to core-shell particles [29]. In our research, we use the Generalized Nonlocal Optical Response (GNOR) theory [30,31], which proved to be an appropriate tool for studying core-shell particles [32]. The GNOR theory takes into account both the presence of a longitudinal field inside the metal and additional boundary conditions at the interfaces between the metal and the dielectric. It was found that accounting for the quantum effect of spatial dispersion leads to a significant decrease in the plasmon resonance amplitude and a shift of its position to the short-wavelength region. All these circumstances can significantly decrease the efficiency of application of the magnetoplasmonic particles in nanomedicine.

We use the GNOR theory within the framework of the Discrete Sources Method (DSM) [33]. DSM is a rigorous semi-analytical surface-oriented method. It is based on the representation of electromagnetic fields using a finite linear combination of lowest-order distributed multipoles [34] satisfying Maxwell’s semi-classical equations, including longitudinal fields inside a metal shell. Thus, the field representations in all areas satisfy the generalized Maxwell system and the infinity conditions. The corresponding discrete sources (DS) amplitudes are determined from the transmission conditions enforced on the interfaces of the core-shell particle, including additional boundary conditions required for the proper determination of the longitudinal field. Compared to the other surface-based methods, the DSM has some theoretical and numerical advantages. It does not require mesh generation or an integration procedure over a particle surface. It provides both near and far fields directly without any additional computational effort. It enables to solve the scattering problem for all external excitations and polarizations at the same time. An exceptional feature of the DSM is that it allows to estimate the real error of the fields obtained by calculating the residual of the fields at the interfaces of the core-shell particle. This provides an opportunity to compute the near fields with a predetermined numerical accuracy.

It is important to emphasize that DSM is a rigorous semi-analytical method that allows one to treat scattering by non-spherical particles in the presence of the nonlocal effect and obtain results with a high accuracy. DSM has been tested many times and can now be used as a reference code. In particular, the extension of the T-matrix method to the case of non-spherical core-shell particles in the presence of a non-local effect was approved using the DSM computer module. The results obtained with the T-matrix code and the DSM module showed a high accuracy agreement (see [35]). The features mentioned above have already made it possible to apply the DSM to the analysis of plasmonic nanostructures by accounting for the spatial dispersion incorporating the GNOR theory. For instance, the DSM has been successfully used for simulating plasmonic dimers with subnanometric gap [36] and plasmonic nanolaser resonator (SPASER) [37].

The paper is organized as following. Section 2 is devoted to the formulation of the boundary value problem of a plane electromagnetic wave scattering by a core-shell magnetoplasmonic particle and a description of the DSM scheme proposed for its solution. Section 3 presents a numerical analysis of the spatial dispersion influence on the absorption cross-section of magnetic core-shell particles. Some concluding remarks summarize the obtained results that can be found in Section 4.

## 2. Problem Statement and Discrete Sources Method

### 2.1. Scattering Problem Statement

Let us consider a core-shell particle with an axis of symmetry *Oz*, entirely located in an unbounded region of space $D_{e}$ (see Figure 1). The core of the particle is denoted as $D_{c}$, and the shell area as $D_{s}$. Let $\partial D_{c}$ be the interface between the core and the shell and $\partial D_{s}$ the external shell surface. We assume that all media in $D_{i},\;i=e,s,c$ are non-magnetic, and their complex permittivities are $\varepsilon_{i},\;i=e,s,c$.

.

The essence of spatial dispersion (nonlocal effect) is that the relation between the electric field E(M) and displacement D(M) is significantly changed. That is, the local relation D(M)=ε(M)E(M) in space is replaced by an integral one as $D(M)=\int \varepsilon (M-M^{\prime })E(M^{\prime })dM^{\prime }$. One of the consequences of the nonlocality is the appearance of longitudinal electromagnetic fields inside the metal [24]. Thus, the electric field inside the metal is no longer to be purely transverse $\left(divE^{T}=0\right)$ and for an adequate description of the ongoing processes, it is necessary to incorporate longitudinal fields $\left(rotE^{L}=0\right)$.

To account for the spatial dispersion in the metal shell we use the GNOR theory [30]. Within the GNOR, Ohm’s law is generalized for the conduction current inside the metal, that is, the following transition occurs
(1)$$J=\sigma E\Rightarrow \xi^{2}grad\left(divJ\right)+J=\sigma E,$$
where *σ* is the conductivity of the metal, and ξ is nonlocal parameter, the so-called correlation length [38]. Therefore, the corresponding equation for the magnetic field in the Maxwell’s system is changed. As it was mentioned above, inside the metal shell, the electric field consists of transverse and longitudinal fields, that is $E_{s}=E_{s}^{T}+E_{s}^{L},$
$divE_{s}^{T}=0,\;rotE_{s}^{L}=0$. It can be shown [35] that these fields inside the shell satisfy the following Helmholtz equations
(2)$$\Delta E^{T}(M)+k_{T}^{2}E^{T}(M)=0,$$
(3)$$\Delta E^{L}(M)+k_{L}^{2}E^{L}(M)=0,$$
where $k_{T}^{2}=k^{2}\varepsilon_{s},\;k_{L}^{2}=\varepsilon_{s}/\xi$ are the transverse and longitudinal wave numbers, and k=ω/c. The correlation length parameter is defined as $\xi^{2}=\varepsilon_{s}\left(\beta^{2}+D\left(\gamma +j\omega \right)\right)/\left(\omega^{2}-j\gamma \omega \right)$, where $\omega_{p}$ is the plasmon frequency of the metal, γ is the damping coefficient, *β* is the hydrodynamic velocity in the plasma associated with the Fermi velocity $v_{F}$ by the relation $\beta^{2}=3/5v_{F}^{2}$, and *D* is the diffusion coefficient of electrons [39].

Let us consider the boundary value scattering problem of linearly polarized plane wave $\left\{E_{0},\;H_{0}\right\}$ propagating at an angle $\pi -\theta_{0}$ to the axis of symmetry *Oz* (see Figure 1). Then, the boundary value problem statement can be written as:

Maxwell equations inside the magnetic core and the external medium
$$\nabla \times H_{i}=jk\varepsilon_{i}E_{i},\;\nabla \times E_{i}=-jkH_{i}\;\;\mathrm{in}\;D_{i},\;i=c,e;$$

Maxwell equations inside the metal shell
$$\nabla \times H_{s}=jk\left(\varepsilon_{s}+\xi^{2}\nabla \nabla \cdot \right)E_{s}(M),\nabla \times E_{s}=-jkH_{s}\;\;\mathrm{in}\;\;D_{s};$$

Transmission conditions for the fields on the interfaces of the core-shell particle including the additional boundary conditions for the normal component of the fields [39]
(4)$$\begin{array}{c} n_{c}\times \left(E_{c}\left(P\right)-E_{s}\left(P\right)\right)=0,\\ n_{c}\times \left(H_{c}\left(P\right)-H_{s}\left(P\right)\right)=0,\\ \varepsilon_{c}n_{c}\cdot E_{c}(P)=\varepsilon_{L}n_{c}\cdot E_{s}(P),\\ P\in \;\partial D_{c}, \end{array}\;\;\begin{array}{c} n_{s}\times \left(E_{s}\left(P\right)-E_{e}\left(P\right)\right)=n_{s}\times E_{0}\left(P\right),\;\\ n_{s}\times \left(H_{s}\left(P\right)-H_{e}\left(P\right)\right)=n_{s}\times H_{0}\left(P\right),\\ \varepsilon_{L}n_{s}\cdot E_{s}(P)=\varepsilon_{e}n_{s}\cdot \left(E_{e}(P)+E_{0}(P)\right),\\ P\in \;\partial D_{s}; \end{array}$$

Silver-Muller radiating conditions for the scattered fields at infinity [40].
$$\underset{r\to \infty }{\lim}\;r\cdot \left(H_{e}\times \frac{r}{r}-\sqrt{\varepsilon_{e}}E_{e}\right)=0,\;r=\left|M\right|\to \infty \;\mathrm{in}\;D_{e}.$$

Here $\left\{E_{e},\;H_{e}\right\}$ is the scattered field in the external medium $D_{e}$, $\left\{E_{c,s},\;H_{c,s}\right\}$ are the total fields in the corresponding domains $D_{c,s}$, $n_{c,s}$ are the unit normals to the surfaces $\partial D_{c,s}$, k=ω/c and the media parameters are chosen so that Imεe=0, Imεc,s≤0, ImεL≤0, where $\varepsilon_{L}=\varepsilon_{s}-\omega_{p}^{2}/\left(j\gamma \omega -\omega^{2}\right)$. The time dependence has the form exp{jω t}. We will assume that the formulated boundary value problem (4) has a unique solution.

### 2.2. Discrete Sources Method

We will construct an approximate solution to the boundary value problem (4) based on the DSM scheme, accounting for the axial symmetry of the core-shell particle and *P/S* polarization of the incident plane wave [36]. The field of a linearly *P*-polarized plane wave has the form
(5)$$\begin{array}{l} E_{0}^{P}=(e_{x}\cos\theta_{0}+e_{z}\sin\theta_{0})\psi_{0}(x,z),\\ H_{0}^{P}=-\sqrt{\varepsilon_{e}}e_{y}\psi_{0}(x,z) \end{array}$$
and for S-polarization can be written as
(6)$$\begin{array}{l} E_{0}^{S}=e_{y}\psi_{0}(x,z),\\ H_{0}^{S}=\sqrt{\varepsilon_{e}}(e_{x}\cos\theta_{0}+e_{z}\sin\theta_{0})\psi_{0}(x,z), \end{array}$$
where $\psi_{0}(x,z)=\exp\left\{-jk_{e}\left(x\sin\theta_{0}-z\cos\theta_{0}\right)\right\}$, and $e_{x},e_{y},e_{z}$ are the unit vectors of the Cartesian coordinate system.

An approximate solution for the transversal fields is presented by means of the following vector potentials
(7)$$A_{mn}^{(1)i}=\{\;Y_{m}^{i}(\eta ,z_{n}^{i})\cos(m+1)\varphi ;-Y_{m}^{i}(\eta ,z_{n}^{i})\sin(m+1)\varphi ;0\},\;A_{mn}^{(2)i}=\{\;Y_{m}^{i}(\eta ,z_{n}^{i})\sin(m+1)\varphi ;Y_{m}^{i}(\eta ,z_{n}^{i})\cos(m+1)\varphi ;\;0\;\},\;A_{n}^{(3)i}=\{0;0;Y_{0}^{i}(\eta ,z_{n}^{i})\},i=c,e,s\pm ,$$
written in a cylindrical coordinate system. The following notations used here are: $Y_{m}^{c}(\eta ,z_{n}^{c})=j_{m}(k_{c}r_{\eta z_{n}^{c}})P_{m}^{m}(\cos\theta_{z_{n}^{c}})\;,$ $\;j_{m}(.)$ is spherical Bessel function, $Y_{m}^{s\pm }(\eta ,z_{n}^{s})=h_{m}^{\left(2,1\right)}(k_{s}r_{\eta z_{n}^{s}})P_{m}^{m}(\cos\theta_{z_{n}^{s}})\;,$ $\;h_{m}^{\left(2,1\right)}(.)$ are spherical Hankel functions, corresponding to “outward” and “inward” waves, $Y_{m}^{e}(\eta ,z_{n}^{e})=h_{m}^{\left(2\right)}(k_{e}r_{\eta z_{n}^{e}})P_{m}^{m}(\cos\theta_{z_{n}^{e}})\;,$ $P_{m}^{m}(\cos\theta_{z_{n}^{i}})={\left(\rho /r_{\eta z_{n}^{i}}\right)}^{m}$, $r_{\eta z_{n}^{i}}^{2}=\rho^{2}+{\left(z-z_{n}^{i}\right)}^{2}$,  η=(ρ,z) , $k_{i}=k\sqrt{\varepsilon_{i}}$, $z_{n}^{i}$ are positions of multipole sources on the axis of rotation, i=c,e,s. Note that the functions $Y_{m}^{i}(\eta ,z_{n}^{i})\exp(m\varphi )$ satisfy the Helmholtz Equation (2).

For *P*-polarization, the longitudinal field is constructed through scalar potentials of the following form [36]
$$\Psi_{mn}^{s\pm }(M)=h_{m+1}^{\left(2,1\right)}(k_{L}^{s}R_{\eta z_{n}^{s}})P_{m+1}^{m+1}(\cos\theta_{z_{n}^{s}})\cos(m+1)\varphi ,\Psi_{n}^{s\pm }(M)=h_{0}^{\left(2,1\right)}(k_{L}^{s}R_{\eta z_{n}^{s}}),$$
which satisfy the Helmholtz Equation (3). Then the DSM approximate solution accepts the form
$$E_{i}^{TN}=\sum_{m=0}^{M}\;\sum_{n=1}^{N_{i}^{m}}\left\{p_{mn}^{i}\frac{j}{k\varepsilon_{i}}\nabla \times \nabla \times A_{mn}^{(1)i}+q_{mn}^{i}\frac{1}{\varepsilon_{i}}\nabla \times A_{mn}^{(2)i}\right\}+\sum_{n=1}^{N_{i}^{0}}r_{n}^{i}\frac{1}{k\varepsilon_{i}}\nabla \times \nabla \times A_{n}^{(3)i},$$
(8)$$E_{\tau }^{LN}=\sum_{m=0}^{M}\sum_{n=1}^{{\bar{N}}_{\tau }^{m}}{\bar{p}}_{mn}^{\tau }\nabla \Psi_{mn}^{\tau }+\sum_{n=1}^{{\bar{N}}_{\tau }^{0}}{\bar{r}}_{n}^{\tau }\nabla \Psi_{n}^{\tau },\;\tau =i,s\pm ,H_{i}^{N}=\frac{j}{k}\nabla \times E_{i}^{N},\;i=e,c,s\pm .$$

Note that inside the shell, the electromagnetic field is constructed as the sum of “outward” and “inward” waves $E_{s}^{TN}=E_{s+}^{TN}+E_{s-}^{TN}+E_{s+}^{LN}+E_{s-}^{LN},\nabla \cdot E_{s\pm }^{TN}=0,\nabla \times E_{s\pm }^{LN}=0$.

In the case of S-polarization, the longitudinal field is presented using the following potentials
$$\Psi_{mn}^{s\pm }(M)=h_{m+1}^{\left(2,1\right)}(k_{L}R_{\eta z_{n}^{s}})P_{m+1}^{m+1}(\cos\theta_{z_{n}^{s}})\sin(m+1)\varphi .$$

In this case the fields are written as
(9)$$E_{i}^{TN}=\sum_{m=0}^{M}\;\sum_{n=1}^{N_{i}^{m}}\left\{p_{mn}^{i}\frac{j}{k\varepsilon_{i}}\nabla \times \nabla \times A_{mn}^{(2)i}+q_{mn}^{i}\frac{1}{\varepsilon_{i}}\nabla \times A_{mn}^{(1)i}\right\}+\sum_{n=1}^{N_{i}^{0}}r_{n}^{i}\frac{1}{\varepsilon_{i}}\nabla \times A_{n}^{(3)i},\;E_{\tau }^{LN}=\sum_{m=0}^{M}\sum_{n=1}^{N_{\tau }^{m}}{\bar{p}}_{mn}^{\tau }\nabla \Psi_{mn}^{\tau },\;\tau =c,s\pm ,H_{i}^{N}=\frac{j}{k}\nabla \times E_{i}^{N},\;i=e,c,s\pm .$$

The approximate solutions (8) and (9) satisfy all the conditions of the boundary value problem, except for the transmission conditions for the fields at the interfaces $\partial D_{i,s}$. The unknown amplitudes of the multipoles $\left\{p_{mn}^{i},\;q_{mn}^{i},\;r_{n}^{i};\;{\bar{p}}_{mn}^{\tau },\;{\bar{r}}_{n}^{\tau }\right\}$ are determined from these transmission conditions (4). For this purpose, we expand the incident plane wave $\psi_{0}(x,z)$ into a Fourier series with respect to the azimuth angle *φ*. In view of the representations (8) and (9), we deduce that the scattering problem on the axial symmetric interfaces $\partial D_{i,s}$ decouples over the azimuth modes *m*, and a separate solution for each mode can be obtained. To satisfy the transmission conditions, we use the generalized point matching technique for the Fourier harmonics of the fields [41] on the surface profiles. Essentially, we are led to an overdetermined system of equations for an amplitude vector of each harmonic and this vector is computed as pseudosolution of the system for all incidences and polarizations at once. More details can be found in [42].

## 3. Computer Simulating Results

Once the amplitudes of the discrete sources are determined, it is easy to compute the *P/S*-polarized scattered fields using representations (8) and (9). In this section we present some numerical results for magnetic gold covered core-shell particles. We will be interested in the analysis of the absorption cross-section, which is responsible for the concentration of electromagnetic energy inside the particle:(10)σabs(θ0,λ)=−Re∫∂Ds(EeN+E0)×(HeN+H0)∗dσ.

Consider the core-shell Fe_n_O_m_@Au particle placed in an ambient medium with a refractive index $n_{e}=\sqrt{\varepsilon_{e}}$. The gold frequency dependent refractive index $n_{s}=\sqrt{\varepsilon_{s}}$ is taken from [43] and the magnetite Fe_n_O_m_ ones from [44]. The corresponding GNOR parameters for gold are used in accordance with [38]:$$\hbar \omega_{p}=9.02\;\mathrm{eV},\hbar \gamma =0.071\;\mathrm{eV},v_{F}=1.39\cdot 10^{12}\;\mu m/s,D=8.62\cdot 10^{8}\;\mu m^{2}/s.$$

We start our research with spherical core-shell particles and estimate the spatial dispersion effect on the position and amplitude of LSPR.

Figure 2a shows the absorption cross-section *σ* versus the exciting wavelength *λ* for spherical core-shell Fe_3_O_4_@Au particle deposited in water (*n_e_* = 1.33) with core diameter *D* = 16 nm and different shell thicknesses *d*. These results are obtained for the local response (LR). One can see that a reduce in the shell thickness leads to a shift of the maximum value to the near infrared region and its decrease. The red curve refers to the case with a larger core diameter *D* = 18 nm. Figure 2b demonstrates the effect of the non-local response (NL) for the same particle as in Figure 2a. The graphs show that accounting for spatial dispersion in the gold shell reduces the maximum values by about 15% accompanied by a blue shift of 10 nm. By a blue shift, we mean the shift of the absorption cross-section curves towards the blue end of the spectrum or high frequencies (short wavelengths). A red shift means a shift of the curves towards longer wavelengths.

An important issue when it comes to using core-shell particles is the requirement for the maximum of the absorption cross section to fit the transparency window of human tissues, which is in the optical range of 700–950 nm [45,46].

In Figure 3a we plot the absorption cross-sections of two particles with different magnetic core materials Fe_3_O_4_ (α) and Fe_2_O_3_ (γ) [15], core diameter *D* = 16 nm, and gold shell thickness *d* = 2 nm for local and non-local cases. The γ-core exhibits a larger maximum value of the absorption cross-section and red shift beyond 800 nm. At the same time, spatial dispersion reduces the maximum by about 25%, and the blue shift exceeds 15 nm.

Figure 3b is related to the examination of the influence of asymmetry in the core-shell structure when the core center is shifted with respect to the center of the shell. We consider two core materials α-γ and the case of a local response. The shift Δ is equal to 0.7 nm. In this case the shell thickness varies from 1.3 nm on one side of the particle to 2.7 nm on the other. The figure shows the results averaged by the angles of incidence and polarizations. It is worth to mention that the averaging procedure is computationally inexpensive since it is performed in one run of the DSM code. This is due to the fact that DSM is a direct method and enables to compute both polarizations and all incidences simultaneously. As expected, the thinner part of the shell shifts the maximum towards longer wavelengths, while decreasing its value.

The influence of the spatial dispersion on the averaged results for the asymmetric core-shell particle is shown in Figure 4a. Two different core materials α-γ are considered. As before, the nonlocal response leads to a decrease in the maxima up to 25%, and the blue shift exceeds 12 nm.

Figure 4b shows the local response results for an ideal core-shell particle and the averaged non-local response results associated with an asymmetric particle. It is interesting to note that the asymmetry red shift is compensated by a blue shift associated with the spatial dispersion.

In Figure 5a we demonstrate the influence of the spatial dispersion on the absorption cross-section curves computed for the symmetric Fe_3_O_4_ core-shell particle with the same gold shell thickness *d* = 2 nm and different core diameters. One can see that for all particle core diameters, nonlocal response curves maxima are about 25% less and shifted to the left by 12 nm with respect to the local response plots.

The results related to particles of different core diameters can be observed in Figure 5a. As expected, an increase in the core diameter with a simultaneous increase in the amount of gilding leads to an increase in the absorbed electro-magnetic energy with a simultaneous shift to longer wavelengths region.

Figure 5b presents the local and nonlocal response results for the symmetric Fe_3_O_4_ particle with core diameter *D* = 16 nm and gold shell thickness *d* = 2 nm deposited in different ambient media: water, human tissue: healthy breast, and stomach [46]. Obviously, denser media give a larger absorption cross-section accompanied by a slight red shift. As before, the spatial dispersion leads to a decrease in the absorption amplitude and a shift to shorter wavelengths.

All computer simulations were performed on Intel core i7-8550U, 1.8 GHz, RAM 8 Gb computer. As an example, one run of the DSM code for asymmetric core-shell particle *D* = 16 nm, *d* = 2 nm in the wavelength range from 650 to 850 nm with 5 nm increment and with the accuracy control enabled took 143 s.

## 4. Discussion

Layered nanostructures appear to be a more convenient tool for manipulating the amplitude and position of plasmon resonance (PR) in the frequency domain than homogeneous ones. In the case of magnetooptics, by changing the diameter of the core and the thickness of the gold shell, it is possible to shift the PR into the transparency window of human tissues, thereby enhancing the therapeutic effect and reducing the risk of damage to healthy tissues. Numerical experiments have shown that by reducing the thickness of the gold shell to 2 nm, it is possible to shift the maximum of the electromagnetic energy absorption to a range beyond 700 nm. However, with such a thickness of the gold shell, it is necessary to take into account the quantum effect of spatial dispersion that occurs in plasmonic metals. Within the framework of the GNOR theory, the DSM was adjusted for studying magnetoplasmonic nanostructures of the magnetit @ Au type. The DSM computer model makes it possible to investigate the optical characteristics of core-shell structures, including asymmetric ones, accounting for the spatial dispersion in plasmonic materials. It is important to emphasize that when studying the influence of asymmetry, the absorption cross section was averaged over both polarizations and directions of external excitation. This seems to be necessary, since it is not known in advance how the particle is positioned relative to the direction of the incident light. After computer simulations, the following main results were obtained:The gold shell thinning leads to a shift of the PR to the infrared region, while simultaneously reducing its amplitude.Spatial dispersion reduces the PR with a simultaneous blue shift of its maximum.The use of core materials Fe_3_O_4_ and Fe_2_O_3_ provides an additional opportunity to shift the PR towards the human tissue transparency window, increasing the amplitude of the energy absorption.The asymmetry of the core-shell particles leads to a decrease in the intensity of the absorbed energy with a shift towards longer wavelengths.It is interesting to note that the spatial dispersion and asymmetry of the particle lead to mutual compensation for the shift in the PR position, causing only a significant decrease in the PR amplitude.An increase in the core diameter causes an increase in energy absorption accompanied by a shift in the PR to the longer wavelengths.The deposition of a particle in denser media leads to a larger absorption cross-section accompanied by a slight red shift.

As a result of the computational experiment, a significant influence of spatial dispersion on the position of the maximum and the amplitude of the absorbed electromagnetic energy was found. It was shown that the magnitude of the decrease can reach 25%, and the shift may exceed 15 nm.

## Figures and Tables

**Figure 1 nanomaterials-11-03297-f001:**
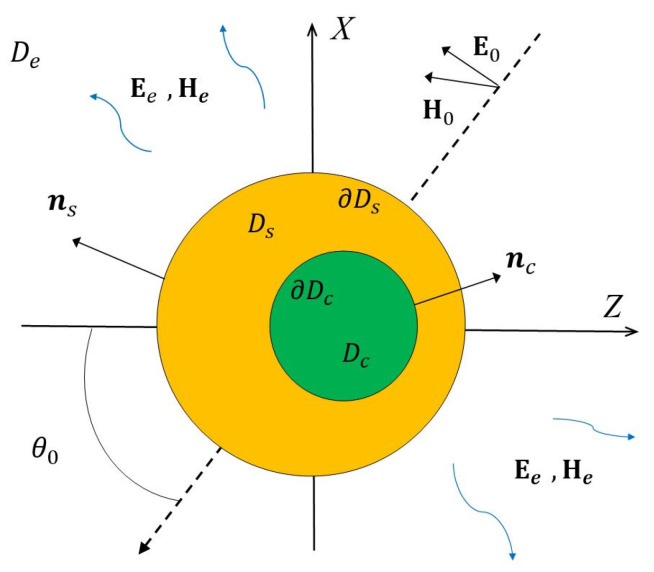
Core-shell particle illuminated by the plane wave $\left\{E_{0},\;H_{0}\right\}$.

**Figure 2 nanomaterials-11-03297-f002:**
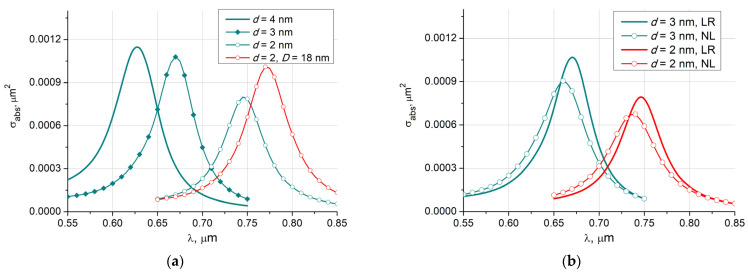
Absorption cross-section versus the exciting wavelength for spherical core-shell Fe_3_O_4_@Au particle with core diameter *D* = 16 nm: (**a**) different shell thicknesses *d*; (**b**) local response (LR), and non-local response (NL) results.

**Figure 3 nanomaterials-11-03297-f003:**
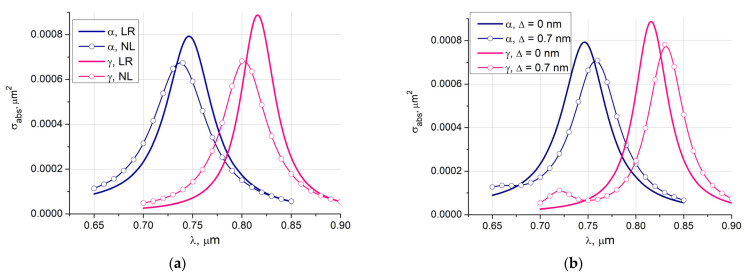
Absorption cross-sections of two particles with different magnetic core materials Fe_3_O_4_ (α) and Fe_2_O_3_ (γ), core diameter *D* = 16 nm, gold shell thickness *d* = 2 nm: (**a**) local response (LR) and non-local response (NL) results; (**b**) influence of the shifted by Δ nm core for the local response. The results are averaged by the angles of incidence and polarizations.

**Figure 4 nanomaterials-11-03297-f004:**
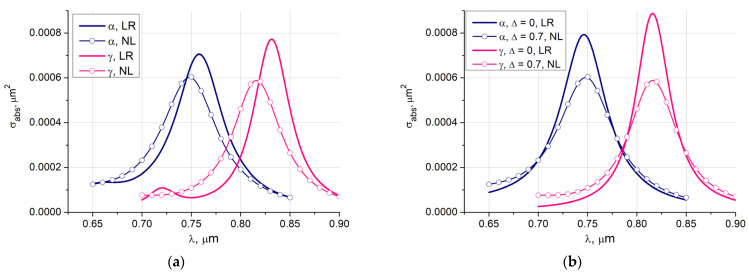
Absorption cross-sections of two particles with different magnetic core materials Fe_3_O_4_ (α) and Fe_2_O_3_ (γ), core diameter *D* = 16 nm, gold shell thickness *d* = 2 nm: (**a**) averaged local response (LR) and non-local response (NL) results for the particle with a shifted by Δ *=* 0.7 nm core; (**b**) local response (LR) curves for symmetrical core-shell particles and the averaged non-local response (LR) results for the particles with shifted cores.

**Figure 5 nanomaterials-11-03297-f005:**
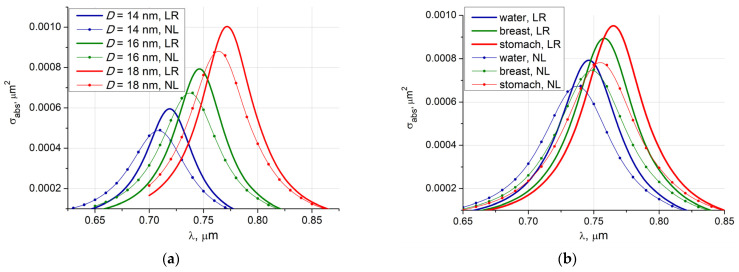
Absorption cross-section versus the exciting wavelength for spherical core-shell Fe_3_O_4_@Au particle with gold shell thickness *d* = 2 nm: (**a**) particle is deposited in water, results for different core diameters *D* and local (LR), non-local responses (NL) are presented; (**b**) particle with core diameter *D =* 16 nm is deposited in different ambient media: water *n_e_* = 1.33, breast *n_e_* = 1.405, stomach *n_e_* = 1.446.

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
