# Peer review of "Influence of Spatial Dispersion on the Electromagnetic Properties of Magnetoplasmonic Nanostructures"

_nanomaterials, 2021, doi:10.3390/nano11123297_

Round 1

Reviewer 1 Report

In this manuscript the Authors consider the role of spatial dispersion on the electromagnetic properties  of magnetoplasmonic nanostructures, particularly relevant for very small-sized nanostructures. The absorption cross-section is numerically evaluated in some relevant cases, and the role of spatial dispersion in the absorption spectra is discussed, as well as the effect of an asymmetry in the core-shell structure.

The research subject is relevant for both basic and applied science (in nanomedicine, in particular).

This paper is scientifically sound, seems relevant for its specific research field, and it is clearly written. 

I would only ask the authors to consider the following points, that in my opinion could contribute to improve the manuscript:

  • At the end of the Abstract, what the phrase "the blue shift can..." refers to should be better specified.
  • In the Introduction, the Authors describe the analytical/numerical methodology they use, but do not summarize their main results. I would suggest to add a paragraph at the end of the Introduction summarazing the main results obtained. In my opinion, this would improve the readability of the paper.
  • The quantity $\epsilon_s$ after Eq. (3) has not been defined. Is it frequency-dependent?

Author Response

Dear Reviewer!

We appreciate your careful consideration of our article. We hope that the corrections and additions made to the article in accordance with your recommendations allowed us to improve the text of the article for better understanding of the results obtained. Below we provide our answers to your comments.

I would only ask the authors to consider the following points, that in my opinion could contribute to improve the manuscript:

  1. At the end of the Abstract, what the phrase "the blue shift can..." refers to should be better specified.

Answer: By blue shift, we mean the shift of the absorption cross-section curves towards the blue end of the spectrum or high frequencies (short wavelengths). A red shift means a shift of the curves towards longer wavelengths.

The above text has been added to the Section 3 «Computer simulating results», when discussing the results presented in Figure 2.

  1. In the Introduction, the Authors describe the analytical/numerical methodology they use, but do not summarize their main results. I would suggest to add a paragraph at the end of the Introduction summarazing the main results obtained. In my opinion, this would improve the readability of the paper.

Answer: In the Introduction we have added some references to recent articles that cover other interesting DSM results and applications. In particular, articles on modeling plasmonic dimers with a subnanometric gap and a plasmonic nanolaser (SPASER) are included.

  1. The quantity $\epsilon_s$ after Eq. (3) has not been defined. Is it frequency-dependent?

Answer: It has been defined in the first paragraph of Section 2.1 together with the other quantities: ””. In the second paragraph of Section 3 after formula (10) we mention its frequency dependence: "The gold frequency dependent refractive index  is taken from [43]…".

Once again, we would like to thank you for all your suggestions.   

Reviewer 2 Report

The authors describe the optical properties of FenOm core and Au shell nanoparticles by GNOR theory, particolarly take into account the quantum effect of spatial dispersion of metal.
In addition asymmetric case of the core shell has been dealt with as well as the effects due to the shell thickness and the core materials.
For these reasons, the paper is consistent and well presented and I suggest  major revisions.

 1) The introduction is poor in literature
 2) the figures have to be improved in resolution
 3) I suggest simulations with different core dimensions
 4) Is it possible to perform simulations in different external environment to simulate human tissue? How change their optical responses?

Author Response

Dear Reviewer!

We appreciate your careful consideration of our article. We hope that the corrections and additions made to the article in accordance with your recommendations allowed us to improve the text of the article for better understanding of the results obtained. Below we provide our answers to your comments.

The Reviewer's suggestions:

  1. The introduction is poor in literature

Answer: We have expanded the Introduction section and added 11 new references. Some of them are review papers.

  1. The figures have to be improved in resolution

Answer: The figures resolution has been improved.

  1. I suggest simulations with different core dimensions.

Answer: We have performed simulations for different core diameters and presented the results in additional Fig. 5a.

  1. Is it possible to perform simulations in different external environment to simulate human tissue? How change their optical responses?

Answer: Simulations related to the deposition of a particle in various external media corresponding to human tissues have been performed. Water, breast tissue and stomach tissue have been considered. The results are shown in Fig. 5b.

Once again, we would like to thank you for all your suggestions.

Reviewer 3 Report

This work systematically studied the electromagnetic properties of magnetoplasmonic nanostructures using DSM model within the framework of the GNOR theory. The results can help us to optimize the design structure and composition of the magnetoplasmonics used for imaging and therapy. The concept and method used are reasonable. I would like to recommend accepting this manuscript.

Author Response

Reviewer report:

This work systematically studied the electromagnetic properties of magnetoplasmonic nanostructures using DSM model within the framework of the GNOR theory. The results can help us to optimize the design structure and composition of the magnetoplasmonics used for imaging and therapy. The concept and method used are reasonable. I would like to recommend accepting this manuscript.

Dear Reviewer, we are grateful for your careful consideration of our article and positive feedback on the obtained results. 

Reviewer 4 Report

This article introduces discrete sources method which incorporates the generalized nonlocal optical response theory to investigate the influence of spatial dispersion on the electromagnetic properties of magnetoplasmonic nanostructures. It was found that taking spatial dispersion into account leads to a decrease in the maximum value of the concentration of electro- magnetic energy up to 25%, while the blue shift can reach 15 nm. The results are interesting enough to potential readers in terms of their optical characteristics. 

1. The research content is novel but not deep enough. Please compare your results with that obtained by using other popular approaches.

2. I would like to note that all obtained results are the numerical result, there would be useful to add a short comment on the way, where the simulations were performed. Moreover, the paper contains only figures encompassing the numerical simulation. An additional drawing with the proposed experimental arrangement would be helpful.

3.  A more important suggestions pertains to the motivation of the work. The influence of spatial dispersion on the electromagnetic properties of magnetoplasmonic nanostructures is presented. To increase the scope and interest of potential readers, I suggest that the authors present more applications where this concept may be useful or interesting to explore for future works.

Author Response

Dear Reviewer!

We appreciate your careful consideration of our article. We hope that the corrections and additions made to the article in accordance with your recommendations allowed us to improve the text of the article for better understanding of the results obtained. Below we provide our answers to your comments.

The Reviewer's suggestions:

  1. The research content is novel but not deep enough. Please compare your results with that obtained by using other popular approaches.

Answer: It is important to emphasize that DSM is a rigorous semi-analytical method that allows one to treat scattering by non-spherical particles in the presence of the nonlocal effect and obtain results with a high accuracy. DSM has been tested many times and can now be used as a reference code. In particular, the extension of the T-matrix method to the case of non-spherical core-shell particles in the presence of a non-local effect was approved using the DSM computer module. The results obtained with the T-matrix code and the DSM module showed a high accuracy agreement (see Fig. 2) [Doicu A. et al.]. Doicu A, Eremin Y, Wriedt T. Transition matrix of a nonspherical layered particle in the non-local optical response theory. J. Quant. Spectr. Radiat. Transfer. 2020. 254. 107196.

The above text has been added to the Introduction as well as a reference to the related article.

  1. I would like to note that all obtained results are the numerical result, there would be useful to add a short comment on the way, where the simulations were performed. Moreover, the paper contains only figures encompassing the numerical simulation. An additional drawing with the proposed experimental arrangement would be helpful.

Answer: All computer simulations were performed on a Lenovo X280 laptop, Intel core i7-8550U, 1.8GHz, RAM 8Gb. As an example, one run of the DSM code for asymmetric core-shell particle D=16nm, d=2nm in the wavelength range from 650 to 850nm with 5nm increment and with the accuracy control enabled took 143sec.

The above text has been added to the end of Section 3 «Computer simulating results».

Answer: In our article, only the simulation results are presented. Unfortunately, we do not have the necessary experience in measurements to propose an appropriate experimental setup.

  1. A more important suggestions pertains to the motivation of the work. The influence of spatial dispersion on the electromagnetic properties of magnetoplasmonic nanostructures is presented. To increase the scope and interest of potential readers, I suggest that the authors present more applications where this concept may be useful or interesting to explore for future works.

Answer: In the introduction we have added some references to recent articles that cover other interesting DSM applications. In particular, articles on modeling plasmonic dimers with a subnanometric gap and a plasmonic nanolaser (SPASER) are included.

Once again, we would like to thank you for all your suggestions. 

Round 2

Reviewer 2 Report

---